# Endogenous PGD$_2$ acting on DP2 receptor counter regulates *Schistosoma mansoni* infection-driven hepatic granulomatous fibrosis

**Giovanna N. Pezzella-Ferreira[1‡], Camila R. R. Pão[1‡], Isaac Bellas[1‡], Tatiana Luna-Gomes[2], Valdirene S. Muniz[3], Ligia A. Paiva[3], Natalia R. T. Amorim[1], Claudio Canetti[1], Patricia T. Bozza[4], Bruno L. Diaz [1‡]*, Christianne Bandeira-Melo[1‡]***

1 Laboratório de Inflamação, Instituto de Biofísica Carlos Chagas Filho, Universidade Federal do Rio de Janeiro, Rio de Janeiro, Brazil, 2 Departamento de Ciências da Natureza, Instituto de Aplicação Fernando Rodrigues da Silveira, Universidade do Estado do Rio de Janeiro, Rio de Janeiro, Brazil, 3 Laboratório de Imunofarmacologia e Inflamação, Instituto de Ciências Biomédicas, Universidade Federal do Rio de Janeiro, Rio de Janeiro, Brazil, 4 Laboratório de Imunofarmacologia, Instituto Oswaldo Cruz, Fundação Oswaldo Cruz, Rio de Janeiro, Brazil

‡ GNP-F, CRRP and IB contributed equally to this work as first authors.
‡ BLD and CBM contributed equally to this work as senior authors.
* bldiaz@biof.ufrj.br (BLD); cbmelo@biof.ufrj.br (CBM)

**Data Availability Statement:** The authors confirm that all data underlying the findings are fully

## Abstract

Identifying new molecular therapies targeted at the severe hepatic fibrosis associated with the granulomatous immune response to *Schistosoma mansoni* infection is essential to reduce fibrosis-related morbidity/mortality in schistosomiasis. *In vitro* cell activation studies suggested the lipid molecule prostaglandin D$_2$ (PGD$_2$) as a potential pro-fibrotic candidate in schistosomal context, although corroboratory *in vivo* evidence is still lacking. Here, to investigate the role of PGD$_2$ and its cognate receptor DP2 *in vivo*, impairment of PGD$_2$ synthesis by HQL-79 (an inhibitor of the H-PGD synthase) or DP2 receptor inhibition by CAY10471 (a selective DP2 antagonist) were used against the fibrotic response of hepatic eosinophilic granulomas of *S. mansoni* infection in mice. Although studies have postulated PGD$_2$ as a fibrogenic molecule, HQL-79 and CAY10471 amplified, rather than attenuated, the fibrotic response within schistosome hepatic granulomas. Both pharmacological strategies increased hepatic deposition of collagen fibers — an unexpected outcome accompanied by further elevation of hepatic levels of the pro-fibrotic cytokines TGF-β and IL-13 in infected animals. In contrast, infection-induced enhanced LTC$_4$ synthesis in the schistosomal liver was reduced after HQL-79 and CAY10471 treatments, and therefore, inversely correlated with collagen production in granulomatous livers. Like PGD$_2$-directed maneuvers, antagonism of cysteinyl leukotriene receptors CysLT1 by MK571 also promoted enhancement of TGF-β and IL-13, indicating a key down-regulatory role for endogenous LTC$_4$ in schistosomiasis-induced liver fibrosis. An ample body of data supports the role of *S. mansoni*-driven DP2-mediated activation of eosinophils as the source of LTC$_4$ during infection, including: (i) HQL-79 and CAY10471 impaired systemic eosinophilia, drastically decreasing eosinophils within peritoneum and hepatic granulomas of infected animals in

available without restriction. All relevant data are within the paper and its Supporting Information files.

**Funding:** This work was supported by Conselho Nacional de Desenvolvimento Científico e Tecnológico (Brazil, grant 310475/2017-1 and 406019/2021-5 to CBM); Fundação de Amparo à Pesquisa do Estado do Rio de Janeiro (grants E-26/202.926/2015 and E26/302.534/2019 to CBM; E-26/203.013/2018 and E-26/201.189/2022 to BLD); and by fellowships from Coordenação de Aperfeiçoamento de Pessoal de Nível Superior (to TLG and NRA) and Conselho Nacional de Desenvolvimento Científico e Tecnológic to GNP-F. The funders had no role in study design, data collection and analysis, decision to publish, or preparation of the manuscript.

**Competing interests:** The authors have declared that no competing interests exist.

parallel to a reduction in cysteinyl leukotrienes levels; (ii) peritoneal eosinophils were identified as the only cells producing LTC$_4$ *in* PGD$_2$-mediated *S. mansoni*-induced infection; (iii) the magnitude of hepatic granulomatous eosinophilia positively correlates with *S. mansoni*-elicited hepatic content of cysteinyl leukotrienes, and (iv) isolated eosinophils from *S. mansoni*-induced hepatic granuloma synthesize LTC$_4$ *in vitro* in a PGD$_2$/DP2 dependent manner. So, our findings uncover that granulomatous stellate cells-derived PGD$_2$ by activating DP2 receptors on eosinophils does stimulate production of anti-fibrogenic cysLTs, which endogenously down-regulates the hepatic fibrogenic process of *S. mansoni* granulomatous reaction — an *in vivo* protective function which demands caution in the future therapeutic attempts in targeting PGD$_2$/DP2 in schistosomiasis.

## Author summary

Accumulation of scar tissue (fibrosis) in the liver is the main cause of health problems associated with schistosomiasis even after the parasite is eliminated by treatment with anthelmintics. Previous experiments with isolated cells in culture have identified a potential role for a lipid mediator, PGD$_2$, in promoting liver fibrosis leading to suggestions that PGD$_2$ inhibition may be beneficial for the people infected with Schistosoma parasite. However, there was no direct evidence in an infection model to support these claims. Here, we described the effect of inhibiting the production or action of PGD$_2$ in a mouse model of schistosomiasis. We identified the cell target and mechanism of action of PGD$_2$'s participation in schistosomiasis. However, our data indicates that PGD$_2$ protects the liver from fibrosis. Thus, inhibition of PGD$_2$ action in patients infected with the Schistosoma parasite may aggravate the condition and promote faster liver failure and should not be pursued as a treatment option.

## Introduction

Often resulting in portal hypertension and liver failure, liver fibrosis is a collateral outcome of hepatic diseases of varying etiologies, such as schistosomiasis — a neglected tropical immuno-pathology, whose fibrosis-driven morbidity and mortality affects millions of people infected with *Schistosoma spp* parasites worldwide. Current anti-schistosomal chemotherapy effectively kills worms resolving the infection with only mild side effects. Yet, controversy persists about whether the present therapeutic approach successfully treats the schistosomiasis-induced liver fibrotic sequelae [1–3].

In *Schistosoma mansoni* (*S. mansoni*) infection-driven pathogenesis, hepatic fibrosis is a long-lasting feature of the excessive granulomatous inflammation built around liver-trapped eggs. The assembly of *S. mansoni*-induced hepatic fibrotic granulomas follows a switch from an initial type 1 to the pro-fibrotic type 2 immune response. And though protective at its onset, the large extensions occupied by the persistent fibrotic granulomatous tissue can culminate with disruption of the normal liver architecture and loss of its essential functions. Initiated by the egg deposition in the liver parenchyma and orchestrated by type 2 cytokines and intense eosinophilic inflammation, the hepatocellular response within schistosomal granulomatous space triggers complex cellular events — remarkably, the transdifferentiation of resident stellate cells into hepatic myofibroblasts [4]. Intra-granulomatous activation of stellate

myofibroblast evokes prominent collagen synthesis and periovular deposition in the *S. mansoni*-infected liver parenchyma [5]. It is well-established that the granulomatous myofibroblast-ruled fibrogenic process is notably promoted by transforming growth factor-β1 (TGF-β1) and IL-13; indeed, the specific blockade of TGF-β1 and IL-13 reduces the schistosomal fibrosis [6,7]. Of note, besides their role in fibrosis as the pivotal collagen-synthesizing cells, *S. mansoni* infection-activated stellate myofibroblasts also appear to display additional immunomodulatory functions through the release of chemokines, cytokines, and bioactive lipid mediators, such as prostaglandin (PG)D$_2$ [8].

PGD$_2$ is a downstream metabolite of the arachidonic acid/cyclooxygenase (COX) enzymatic pathway, which is synthesized by the rate-limiting PGD synthase (PGDS) enzymes. In *S. mansoni* mammal hosts, in addition to the lipocalin-type PGDS highly expressed in the central nervous system, the hematopoietic PGD synthase (hPGDS) is the main terminal enzyme for PGD$_2$ synthesis in peripheral tissues [9]. Besides eosinophils and various inflammatory cell types, hepatic stellate myofibroblasts of the *S. mansoni*-driven granuloma express hPGDS and are notable sources of PGD$_2$ [8–11].

Either autocrinally or paracrinally, PGD$_2$ exerts most of its effects through the activation of two distinct 7-transmembrane G-protein coupled receptors, DP1 and DP2, co-expressed in cell surfaces. In contrast to the adenylate cyclase-driven inhibitory feature of DP1 activation, DP2 (formerly known as CRTH2) elicits the distinctive downstream signaling classically displayed by chemoattractant receptors. The resultant of concomitant activating both PGD$_2$ receptors is a cell type-specific phenomenon [12–14]. In eosinophils, for induction of lipid body-compartmentalized leukotriene (LT)C$_4$ synthesis, PGD$_2$-induced concurrent DP1/DP2 activation converges toward triggering cooperative signaling and intracellular events [12,15,16]: DP1 brings about the assembly of new active lipid body compartments and DP2 activates LTC$_4$ synthesizing machinery within newly formed lipid bodies [15,17].

The link between PGD$_2$ and schistosomal immunopathology began 40 years ago with the observation that *S. mansoni* cercariae can generate PGD$_2$ by themselves [18] and has advanced to unveil that (i) not only cercariae but also other *S. mansoni* life stages (remarkably eggs) produce PGD$_2$ via the enzymatic activity of a schistosomal PGDS — the Sm28GST, a 28 kDa glutathione-S-transferase [19–21]; (ii) cercariae-derived PGD$_2$ within infected skin elicits cutaneous immune evasion, decreasing the migratory capability of Langerhans cells towards draining lymph nodes [22]; and (iii) by employing a DP1 receptor-deficient mouse, activation of host DP1 receptor during *S. mansoni* infection was shown to be crucial for the establishment of early type 1 immune response (until 1 week) as well as late worm burden (7 weeks) in the liver [19]. Altogether, the data clearly show PGD$_2$'s role in schistosomal immunopathogenesis and its potential as a novel adjuvant target for the current chemotherapy, focusing on quelling the deleterious impact of parasitic infection on the host [23].

Understanding the role of PGD$_2$ after the host has been parasitized and before liver fibrosis has fully developed may generate medically relevant insight into disease management. Such studies are particularly relevant due to some divergent evidence for the role of PGD$_2$ in fibrotic processes that affect other organs' fibrogenesis. In renal settings, while *in vitro* PGD$_2$ played an anti-fibrogenic role by suppressing the induction of fibrotic phenotype in cultured kidney cells [24,25]; *in vivo*, PGD$_2$ emerged as pro-fibrogenic since DP2 antagonism downregulates renal fibrosis in a chronic model of kidney inflammation [26]. In the lung, while studies on non-asthmatic models of pulmonary fibrosis acknowledge PGD$_2$ itself (via its DP2 receptor) as either a beneficial [27–30] or a deleterious [31] signal against fibrotic development; in the asthmatic lung, however, PGD$_2$/DP2 displays undisputable pro-fibrogenic impact [32,33]. Although simple extrapolation of PGD$_2$'s role in liver fibrosis based on evidence gathered on different tissues may not be straightforward, it is reasonable to speculate that schistosomiasis-

driven fibrosis may be promoted by PGD$_2$, in a similar pro-fibrotic fashion as it is in asthma due to the shared type 2-governed eosinophilic environment.

Since PGD$_2$ elusive function on fibrogenesis has not been properly validated using *in vivo* models of hepatic inflammation, the potential link between PGD$_2$ and schistosomal hepatic fibrosis is mostly drawn from *in vitro* studies. While by employing DP1 deficient mice, no impact was observed in the fibrotic feature of hepatic granulomatous reactions around *S. mansoni* eggs [19], it has been observed that (i) host PGD$_2$ synthesized by *S. mansoni*-driven stellate myofibroblasts in response to *in vitro* TGF-β autocrinally mediates myofibroblastic secretory activity [34]; (ii) *S. mansoni*-derived PGD$_2$ acting via DP1 triggers *in vitro* eosinophil activation (lipid body biogenesis) [21] and (iii) induces 15-lypoxygenase-driven synthesis of eoxin C$_4$, which in turn promotes TGF-β secretion [21]. Taken together with the pro-fibrotic role in type 2-biased asthma, these pieces of evidence build a case for a PGD$_2$ role in promoting *S. mansoni*-driven fibrogenesis in hepatic granulomas. Moreover, such findings support recommendations of PGD$_2$ as a possible target for anti-fibrotic therapies in schistosomiasis [35–37].

Our investigation of the role of PGD$_2$ in schistosomal granulomas led to the discovery of unanticipated endogenous counter-regulatory functions required to limit hepatic fibrosis beyond those associated with DP1 receptor activation [19,38,39]. We have unveiled a PGD$_2$/DP2 modulatory mechanism that reduces the granuloma-associated fibrotic reaction elicited by *S. mansoni* infection. Although missing pieces of information do not allow the drawing of a complete picture of the events responsible for PGD$_2$-mediated inhibition of *S. mansoni* infection-elicited hepatic granulomatous fibrosis, our findings unveil that once oviposition starts, PGD$_2$ (i) is produced within hepatic granuloma, seemingly, by hPGDS-expressing stellate cells; (ii) activates DP2 receptors to mobilize blood eosinophils to the hepatic granulomas; and (iii) stimulates granuloma-infiltrating eosinophils to synthesize and secrete LTC$_4$; which, via CysLT1 receptor activation (iv) reduces hepatic production of fibrogenic TGF-β and IL-13; (v) decreasing synthesis and deposition of collagen around granuloma-trapped eggs.

## Results and discussion

### *S. mansoni* infection-induced hepatic granuloma formation triggers hPGDS-driven PGD$_2$ synthesis

*In vitro* studies have suggested PGD$_2$ as a potential molecular target for novel anti-fibrotic therapies in the hepatic granulomatous pathology of *S. mansoni* infection [8,35,37]. To address such inference from *in vitro* data, here we first evaluate whether enhanced PGD$_2$ production is elicited in an *in vivo* experimental model of schistosomiasis — C57BL/6 mice infected percutaneously with 60 cercariae of *S. mansoni* (**Fig 1A**). As shown in **Fig 1B**, the infection protocol triggers productive oviposition first detected in feces within 6 weeks, which peaks within 8 weeks post-infection (wpi), while no *S. mansoni* egg was found in feces as early as 3 weeks. Before oviposition (3 wpi), no increase in PGD$_2$ synthesis was detected within peritoneal or hepatic compartments of *S. mansoni*-infected mice (**Fig 1C and 1D**). However, later at 8 wpi, PGD$_2$ levels were elevated in both egg-containing granulomatous liver tissue, as well as, in peritoneal cavities of *S. mansoni*-infected mice (**Fig 1C and 1D**). Of note, we have previously demonstrated that parallel to the 8 weeks-related intense egg release in feces (**Fig 1B**) uncovered here, a type 2 immune response characterized by systemic eosinophilia is associated with the formation of egg-encasing hepatic eosinophilic granulomas with robust fibrosis around *S. mansoni* eggs [40,41]. Although regulatory roles for PGD$_2$ in *S. mansoni*-induced type 2 immune response-biased pathology had been previously observed in functional studies employing the DP1 receptor-deficient mice [19], enhanced PGD$_2$ production in the liver during schistosomiasis had not been demonstrated before.

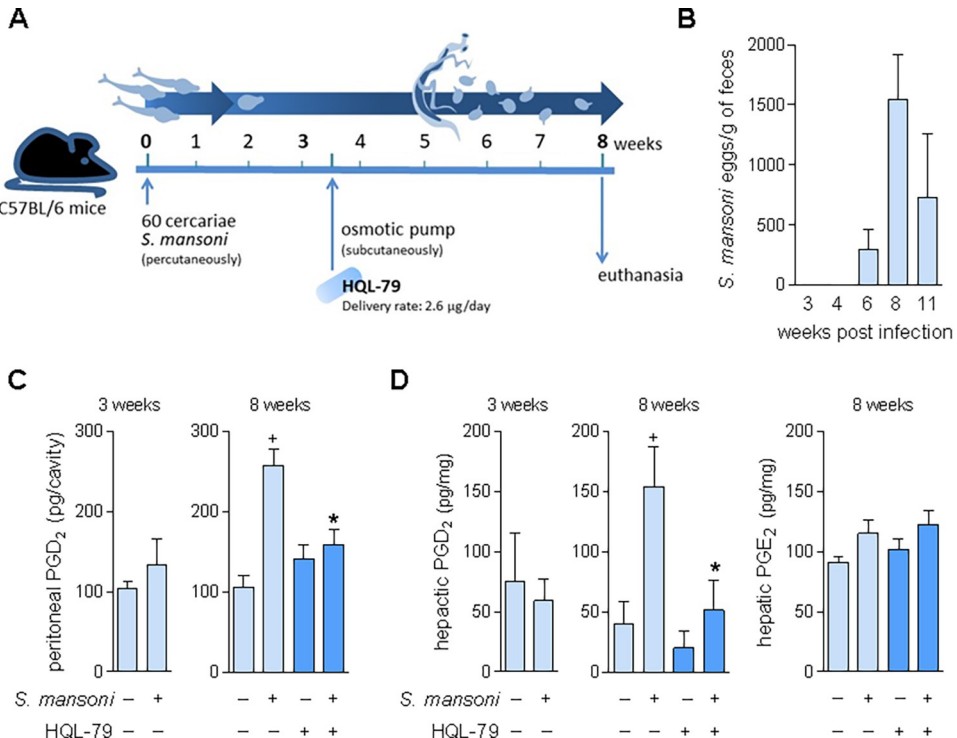

**Fig 1. Characterization of a post-oviposition and host PGD synthase-mediated PGD₂ synthesis during *S. mansoni* infection in mice.** Scheme in **A** outlines the protocol of *S. mansoni* infection in mice achieved by active percutaneous penetration by 60 cercariae and the continuous treatment with HQL-79 (2.6 µg/day), which was delivered by subcutaneously implanted osmotic pumps at 3.5 weeks post infection (wpi) with *S. mansoni*. Peritoneal lavages and livers were collected 3 or 8 wpi. **B** shows a temporal kinetics of *S. mansoni* egg detection in feces. In **C**, PGD₂ amounts detected by specific EIA kit in cell-free peritoneal fluids. In **D,** PGD₂ or PGE₂ levels in liver homogenates detected by specific EIA kits for each prostanoid. Values are expressed as the mean ± SEM from at least 5 animals *per* group (experiments were repeated at least once). $^+p < 0.05$ compared to non-infected control group. $^*p < 0.05$ compared to infected non-treated group.

Administration of HQL-79 — a specific inhibitor of the hPGDS — effectively inhibited the host PGD₂ synthesis induced by *in vivo S. mansoni* infection, thus confirming its adequacy as a pharmacological tool for our aims. As schematized in **Fig 1A**, to achieve systemic and persistent delivery of HQL-79 treatment during long-lasting *S. mansoni* infection, osmotic pumps containing HQL-79 (flow delivery rate of 2.65 µg/day; for about 30 days) were subcutaneously implanted at 3.5 wpi — therefore, before either oviposition (**Fig 1B**) or the onset of *S. mansoni* infection-induced increase of systemic PGD₂ levels (**Fig 1C and 1D**). Validating the systemic reach of this administration strategy, as well the HQL-79 inhibitory specificity towards PGD₂ synthesis, HQL-79 treatment did impair *S. mansoni* infection-induced PGD₂ synthesis detected in both peritoneal cavity and liver (**Fig 1C and 1D**), while did not alter hepatic tissue PGE₂ levels (**Fig 1D**).

## Inhibition of PGD₂ synthesis or DP2 antagonism aggravates fibrosis of *S. mansoni*-induced hepatic granulomas

The proposed pro-fibrogenic function of endogenous PGD₂ in hepatic granulomatous response to *S. mansoni* infection [35,37] was investigated by employing two complementary pharmacological strategies, including either (i) the proven *in vivo* PGD₂ synthesis inhibition by HQL-79 treatment (**Fig 1C** and **1D**); or (ii) the blockade of PGD₂ receptor DP2 by a

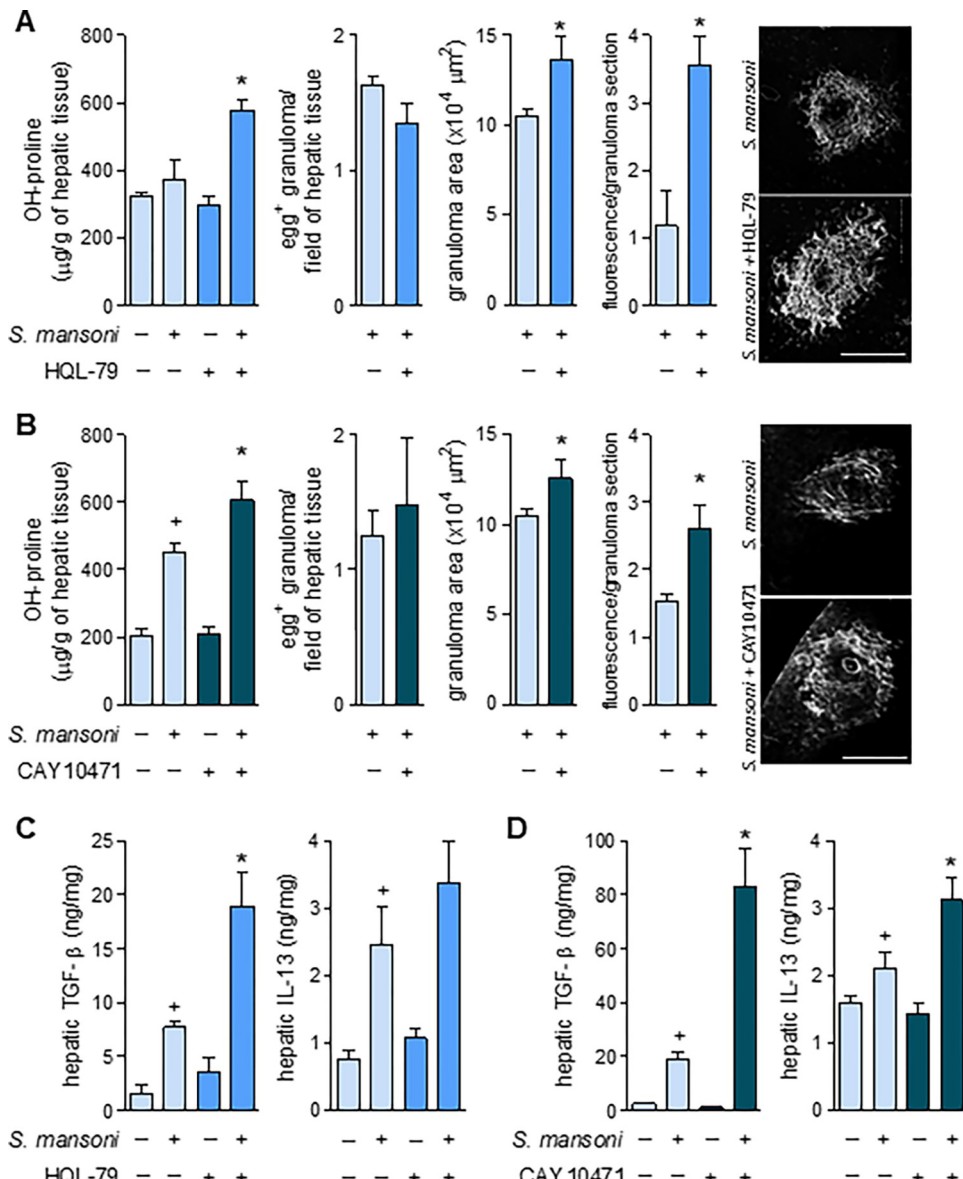

**Fig 2. Inhibition of PGD₂ synthesis by HQL-79 or selective antagonism of DP2 receptor by CAY10471 promote amplification of overall fibrotic reaction within *S. mansoni*-infected hepatic tissue.** Continuous treatments (delivered by implanted osmotic pumps) with either HQL-79 (2.6 μg/day) or CAY10471 (1.7 μg/day) were initiated at 3.5 wpi with *S. mansoni*. Livers were collected 8 wpi. **A** and **B** show total amounts of hepatic hydroxyproline, total numbers of egg-encasing hepatic granulomas, and area measurements of individual hepatic granulomas. Images are representative collagen-related fluorescence in hepatic granuloma sections (confocal microscopy; PicroSirius red modified staining). **C** and **D** show levels of TGF-β and IL-13 detected by specific ELISA kits in liver homogenates of *S. mansoni*-infected mice. Values are expressed as the mean ± SEM from at least 5 animals *per* group from representative experiments for each drug studied. +$p < 0.05$ compared to non-infected control group. *$p < 0.05$ compared to infected non-treated group.

selective antagonist, CAY10471, also delivered by osmotic pump system (flow rate of 1.75 μg/day; implanted at 3.5 wpi). As shown in **Fig 2**, *S. mansoni* infection triggered an intense periovular granulomatous response in livers, presenting enhanced levels of TGF-β and IL-13, as well as locally deposited collagen around *S. mansoni* eggs entrapped in hepatic granulomas. In contrast to the hypothesis of a pro-fibrogenic role of PGD₂ in schistosomiasis derived from *in*

*vitro* studies, in our experiments, both PGD$_2$-targeting treatments aggravated, rather than attenuated, the *S. mansoni*-induced fibrotic reaction in the granulomatous liver. Such unexpected outcome was evidenced by a variety of fibrosis-related hepatic parameters, which were further enhanced by HQL-79 and CAY10471 treatments (**Fig 2**), including *S. mansoni* infection-induced: (i) increased hepatic collagen synthesis (as assessed by total hydroxyproline liver content) (**Fig 2A and 2B**); (ii) deposition of collagen fibers around *S. mansoni* eggs in hepatic granulomas (**Fig 2A and 2B**); (iii) hepatic production of the key pro-fibrogenic cytokine TGF-β (**Fig 2C and 2D**); and (iv) hepatic levels of the prototypical type 2 immune response cytokine IL-13 (**Fig 2C and 2D**) which is known to display potent fibrogenic activity in liver as well [6,7]. Either HQL-79 (**Fig 2A**) or CAY10471 (**Fig 2B**) seemed to worsen overall hepatic *S. mansoni* fibrosis by specifically increasing localized collagen synthesis within granulomas which were already in development; since both treatments: failed to modify the overall number of egg-encasing granulomas found 8 wpi within *S. mansoni*-infected livers, while promoting evident expansion of fibrotic area of individual *S. mansoni*-driven granulomas (**Fig 2A and 2B**). Notably, these findings also indicate that DP2 receptors activation by endogenous PGD$_2$ production — that occurs between 3.5 and 8 wpi — does not play key roles in controlling total egg production or the hepatic burden of parasites. It is important to highlight that our experimental design does not allow general assumptions on whether DP2 activation has endogenous roles in the initial cutaneous phase, oviposition-onset, or granuloma assembly. It is noteworthy that, inversely, DP1 activation by PGD$_2$ during *S. mansoni* infection appears to negatively impact early parasite survival and the successful establishment of infection [19]; a caveat being that DP1 role on *S. mansoni* parasitism is derived from studies with genetically modified mice [19]. Based on classical opposing effects typically displayed by DP1 *versus* DP2 activation [12,42], selective antagonism of DP1 receptor could even trigger beneficial anti-fibrogenic outcomes.

The unexpected pro-fibrotic effect of both HQL-79 and CAY10471 treatments was accompanied by a further increase of the infection-elicited hepatic levels of the pro-fibrogenic factors, TGF-β and IL-13 (**Fig 2C** and **2D**). So, our findings also indicate that the mechanisms involved in the PGD$_2$/DP2-driven down-regulatory function may depend on counterbalancing *S. mansoni*-induced hepatic production of TGF-β and IL-13 *in vivo*. Strikingly, as already mentioned previously, *in vitro* studies showed a diametrically opposite pattern for PGD$_2$ impact on TGF-β or IL-13 secretion by isolated cells, for instance, schistosomal granuloma-derived hepatic stellate cells [21,34,43]. The *in vitro versus in vivo* discrepancies unveiled here for the role of PGD$_2$ in *S. mansoni*-driven fibrotic process are likely due to the single cell type feature of *in vitro* assays, which lacks the sequential cellular activities performed by multiple cell types (*vide infra*) working in the more complex *in vivo* setting of hepatic granulomatous reaction. Of note, besides their significant role in the establishment of fibrosis [4,44], these *S. mansoni*-driven hepatic granulomatous myofibroblasts are also recognized to contribute to local eosinophilia of hepatic periovular granuloma [45].

## Endogenous PGD$_2$ mediates *S. mansoni* infection-elicited eosinophilic inflammation

Eosinophils have recently emerged as protective agents in hepatic tissue, acting towards reinstating liver homeostasis after diverse types of injuries [46,47]. Hepatic granulomatous eosinophilia is an undisputable hallmark of *S. mansoni* infection, yet these cells are considered at most minor immunomodulators of the protective type 2 immune response or even simple bystanders of the pathogenesis [48–50]. On the other hand, it is well-established that eosinophils represent key players in PGD$_2$-regulated inflammatory responses since they function as

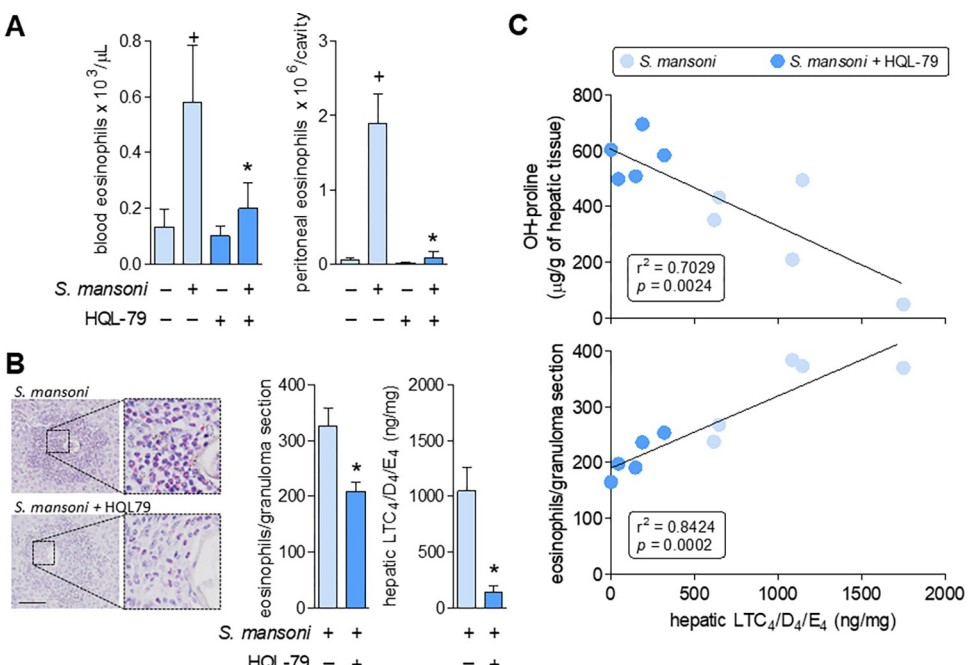

**Fig 3. HQL-79 inhibits eosinophilic reaction and cysLTs production induced by *S. mansoni* infection. A** shows eosinophil numbers found in either peripheral blood or peritoneal compartments 8 wpi with *S. mansoni*. **B** highlights the eosinophilic feature of hepatic granulomas (stained by Sirius red), showing representative images and enumeration of granuloma-infiltrating eosinophils as well as the amounts of cysLTs (LTC$_4$/D$_4$/E$_4$) detected by specific ELISA kits in liver homogenates. **C** displays linear regression curves illustrating the relationships between cysLTS levels and either collagen synthesis (top panel) or granulomatous eosinophilic reaction (bottom panel) found in hepatic tissues. Values are expressed as the mean ± SEM from at least 5 animals *per* group (experiment was repeated at least once). $^+p < 0.05$ compared to non-infected control group. $^*p < 0.05$ compared to infected non-treated group. R-squared (r$^2$) and $p$ values for each linear regression are shown in each panel.

both cellular sources and targets of PGD$_2$ [11,12,15]. With an experimental design not aiming to re-examine eosinophil's role in *S. mansoni*-driven immunopathology, here HQL-79 effect on *S. mansoni* infection-elicited hepatic eosinophilic reaction was examined seeking novel insights on the molecular/cellular mechanisms involved in the unexpected anti-fibrogenic effect of PGD$_2$.

Reproducing clinical schistosomiasis, the mouse experimental model of *S. mansoni* infection used here exhibits, within 8 wpi, a marked systemic eosinophilia characterized by elevated numbers of circulating eosinophils, as well as infiltrating eosinophils found in both peritoneal cavities and in egg-encasing hepatic granulomas (**Fig 3**). Eosinophil recruitment from the circulation into *S. mansoni* infection-elicited inflammatory sites is known to be mediated by a variety of parasite- and host-derived chemoattractant molecules, especially host CCL11 [51]. Based on the potent eosinophilotactic activity of PGD$_2$ [14,52], we analyzed whether endogenous PGD$_2$ would also contribute to *S. mansoni* infection-induced eosinophilic reaction. As shown in **Fig 3**, by systemically inhibiting host PGD$_2$ synthesis, HQL-79 was capable to impair the establishment of *S. mansoni*-driven eosinophilia, decreasing blood eosinophil availability (**Fig 3A, left panel**), eosinophil migration to peritoneal cavities (**Fig 3A, right panel**) and eosinophil influx into *S. mansoni*-induced hepatic granulomas (**Fig 3B, left and middle panels**). Hence, our findings boost the status of hPGDS-derived PGD$_2$ to an important host-derived mediator of *S. mansoni* infection-elicited systemic eosinophilia.

The clear inhibitory effect of HQL-79 towards the eosinophilic feature of *S. mansoni*-driven inflammatory response indirectly uncovers at least three mechanistic aspects of PGD$_2$-mediated regulation of the fibrogenic process of schistosomal hepatic granulomas.

First, HQL-79-unveiled PGD$_2$ role in eosinophilia induced by *S. mansoni* infection is neither a consequence of *S. mansoni* parasitism inhibition nor a failure to mount the subsequent type 2 immune response. Therefore, the mechanism involved is distinct from the observed in DP1 receptor-deficient model [19]. The inhibition of PGD$_2$ synthesis by HQL-79 diminished *S. mansoni* infection-triggered eosinophilic reaction, although the magnitude of parasitic burden (number of eggs trapped in hepatic tissue) was unaffected (**Fig 2A**) and associated with a clear type 2 immune environment. Indeed, besides the slightly elevated hepatic IL-13 levels (although not statistically significant) found under HQL-79 treatment (**Fig 2C**), the PGD$_2$ synthesis inhibitor failed to change the increased serum levels of type 2 cytokines IL-5 and IL-13 (**S1A Fig**) as well as *S. mansoni* infection-triggered eosinophil production at mouse bone marrows (**S1B Fig**), a known IL-5-driven phenomenon.

Second, although well-known producers of PGD$_2$ [10], at first glance granulomatous eosinophils did not seem to be the main host cell producers of PGD$_2$ at hepatic inflammatory site during *S. mansoni* infection, since: (i) HQL-79-promoted reduction of PGD$_2$ derails the establishment of hepatic granulomatous eosinophilia (**Fig 3B**), and more importantly (ii) *S. mansoni* infection-induced hepatic PGD$_2$ synthesis (as analyzed at 6 wpi) precedes eosinophil appearance at the granulomatous reaction sites (**S2 Fig**), occurring despite eosinophils absence. Therefore, the main source for PGD$_2$ synthesis within schistosomal granuloma must be a hPGDS-expressing cell type which is structurally present at schistosomal granuloma at 6 wpi and before the PGD$_2$-mediated eosinophil arrival at hepatic granulomatous tissue. We have previously shown that the *S. mansoni*-driven hepatic stellate cells (granulomatous myofibroblasts) are capable to produce PGD$_2$ in the schistosomiasis context in an HQL-79-sensitive manner [8]. Here, as shown in **S2E Fig**, stellate myofibroblasts purified from 6 wpi hepatic granulomas (for cell isolation methods see [34]), when stimulated *in vitro* with a total lipid extract of *S. mansoni* parasites (for lipid extraction method see [21]), rapidly synthesize and secrete PGD$_2$ within 2 h. Inasmuch as *S. mansoni*-driven myofibroblasts are also known to regulate eosinophilic reaction [45], we can hypothesize that hPGDS-expressing stellate cells are targeted by HQL-79, culminating with both inhibition of hepatic eosinophilia and augmentation of fibrosis-related parameters.

Third, eosinophil does not seem to be the cell source of TGF-β or IL-13 in *S. mansoni*-infected livers. Even though eosinophils are known cell reservoirs of preformed TGF-β and IL-13 [48,53], HQL-79 decreased eosinophil presence (**Fig 3B**), whilst increasing the fibrogenic cytokines (**Fig 2C**) in *S. mansoni*-driven hepatic granulomas. Such inverse relationship indicates that, once in place within hepatic granulomas, eosinophils may suppress, rather than secrete TGF-β and IL-13 themselves or further their release by other granuloma cell types. At the granulomatous microenvironment, a paracrine downregulatory impact, likely orchestrated by secretory activities of PGD$_2$-stimulated eosinophils, may indeed ensure the lower release of TGF-β and IL-13 by other cell sources.

## PGD$_2$ elicits LTC$_4$ production during *S. mansoni* infection: role of eosinophils as LTC$_4$ synthesizing cells

The lipid molecules LTC$_4$, LTD$_4$, and LTE$_4$, collectively known as cysteinyl leukotrienes (cysLTs) are notoriously pro-fibrogenic mediators in a variety of lung conditions [54–56] and similar pro-fibrogenic activities for cysLTs can be anticipated in other organs. However, for hepatic fibrosis of any etiology, the role of these molecules is yet uncharacterized; a notable

exception is a study performed by Toffoli and coworkers (2006) showing that 5-LO-derived metabolites of unknown cellular origin seem to act as suppressing signals of hepatic granulomatous fibrosis in *S. mansoni* infection [57,58]. Here, HQL-79 treatment decreased total amounts of cysLTs found in *S. mansoni*-infected hepatic tissue (**Fig 3B**). Concurring with a cause/effect relationship, and therefore a likely anti-fibrogenic function for these lipid molecules in schistosomal granuloma of infected livers, **Fig 3C** (top panel) shows a significative inverse correlation ($r^2 = 0.7029$; $p \leq 0.05$) between hepatic levels of cysLTs and the fibrosis marker hydroxyproline.

S. mansoni infection-induced egg-entrapping hepatic granulomas are comprised of various functionally active cells, including hepatic stellate myofibroblasts, mast cells, macrophages, and eosinophils [41,51,59]. In addition to *S. mansoni* eggs themselves, all these granuloma-associated cells present the ability to synthesize LTC$_4$ [60]. However, cysLTs production is not a ubiquitous cellular activity, it is rather a highly regulated, stimulus-specific, and cell-restricted phenomenon dependent on, for instance, cellular expression and proper intracellular localization of the limiting LTC$_4$ synthase enzyme. Eosinophils express the entire LTC$_4$ synthesizing enzymatic machinery [61,62], which is promptly activated and compartmentalized within cytoplasmic lipid bodies under PGD$_2$ stimulation [15]. Therefore, at this point, we investigated whether (i) *S. mansoni* infection-induced hepatic LTC$_4$ synthesis may take place within PGD$_2$-stimulated eosinophils infiltrating hepatic granuloma (*vide infra*); and (ii) the reduction of hepatic cysLTs observed under HQL-79 may be in part due to the decreased numbers of LTC$_4$-synthesizing eosinophils within hepatic granulomas. Both assumptions are strongly supported by the significative positive correlation ($r^2 = 0.8424$; $p \leq 0.05$) between the magnitude of eosinophil presence and the levels of LTC$_4$ found in each granuloma-enriched hepatic tissue of *S. mansoni* infected mice, treated or not with HQL-79 (**Fig 3C**; bottom panel). Accordingly, analysis of an early time point (6 wpi) in the *S. mansoni* infection-driven hepatic granuloma development revealed that, before eosinophil infiltration, the hepatic granulomatous reaction does not show LTC$_4$ synthesis (**S2D Fig**). So far, infiltrating eosinophils appear to synthesize/release LTC$_4$ under *in situ* stimulation by granulomatous PGD$_2$ and may represent the main responsible for the 8 wpi increased hepatic levels of cysLTs. Whether activation of eosinophil DP2 receptors mediates this phenomenon was investigated next.

## Production of LTC$_4$ by *S. mansoni* infection-driven eosinophils upon PGD$_2$ activation of DP2 receptors

Besides its eosinophilotactic activity, DP2 receptor is a required element for the successful induction of LTC$_4$ synthesis by PGD$_2$ in eosinophils — a cooperative intracellular process that demands simultaneous activation of both DP1 and DP2 receptors and takes place within the discrete cytoplasmic lipid body organelles [11,12,15,17]. As shown in **Fig 4** and exactly like the inhibition of PGD$_2$ synthesis by HQL-79, the selective antagonism of DP2 receptor by CAY10471 did decrease the number of infiltrating eosinophils found 8 wpi within the peritoneal cavity of *S. mansoni*-infected mice (**Fig 4A**). Infiltrating peritoneal eosinophils of non-treated infected mice displayed high numbers of cytoplasmic lipid bodies (**Fig 4B**) as well as showed intracellular immunolabelling for newly synthesized LTC$_4$ (**Fig 4C** and **4D**). Antagonism of DP2 by CAY10471 diminished the cytoplasmic content of lipid bodies within infiltrating peritoneal eosinophils (**Fig 4B**), shut down the ability of some peritoneal infiltrating eosinophils to synthesize LTC$_4$ (**Fig 4C** and **4D**), and therefore, partially impaired the overall peritoneal production of cysLTs (**Fig 4E**) found 8 wpi with *S. mansoni*. In CAY10471-treated mice, residual peritoneal eosinophilia is still detected (**Fig 4A**) exhibiting a reduced (38.6 ± 6% of inhibition; n = 5, $p \leq 0.05$) LTC$_4$ synthesizing ability (**Fig 4D** and **4E**). Together these

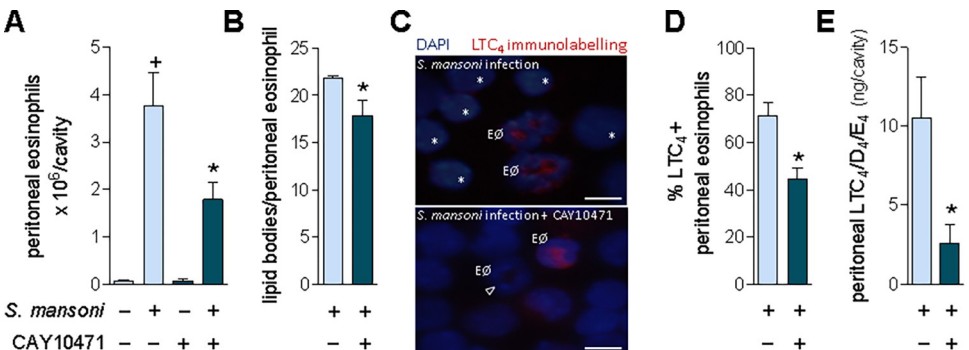

**Fig 4. *S. mansoni* infection induces a PGD$_2$/DP2-mediated cysLTs production by infiltrating peritoneal eosinophils. A** shows peritoneal eosinophil counts while **B** displays the numbers of cytoplasmic lipid body organelles found into peritoneal eosinophils 8 wpi with *S. mansoni*. **C** displays representative images of EicosaCell preparations showing intracellular immunolabelled LTC$_4$ (in red; LTC4$^+$ eosinophils). Cellular nuclei are labelled with DAPI (in blue); bar = 10 μm; "EØ" identifies eosinophils among peritoneal cellular population; arrowhead indicates a LTC$_4^-$ eosinophil (remaining LTC$_4^-$ cells are mononuclear cells indicated by asterisks). In **D**, the percentage of peritoneal eosinophil population exhibiting cytoplasmic red immunostaining for LTC$_4$ (LTC$_4^+$ eosinophils) is shown. **E** shows the total amounts of cysLTs (LTC$_4$/D$_4$/E$_4$) detected by specific ELISA kit in peritoneal fluid supernatants. Values are expressed as the mean ± SEM from at least 5 animals *per* group (each experiment was repeated at least once). $^+p < 0.05$ compared to non-infected control group. $^*p < 0.05$ compared to infected non-treated group.

findings identify eosinophils as, at least, one of the cell types actively participating in *in vivo* PGD$_2$/DP2 receptor-driven stimulation of LTC$_4$ synthesis during *S. mansoni* infection.

As depicted in **Figs 4C** and **S3**, the peritoneal cell population found 8 wpi is not formed only by eosinophils, but by a mixed cell population of eosinophils and mononuclear cells. However, distinct from infiltrating eosinophils, the *S. mansoni* infection-elicited peritoneal mononuclear cells did not display LTC$_4$ synthesizing activity (**Fig 4C** top image; LTC$_4^-$ mononuclear cells indicated by asterisks). In fact, DP2 receptors appear to regulate neither migration nor activation of peritoneal mononuclear cells in this mice model of *S. mansoni* infection. CAY10471 treatment did not alter *S. mansoni* infection-associated numbers of peritoneal mononuclear cells (mixture of mast cells, monocytes/macrophages and lymphocytes) found 8 wpi (**S3A Fig**), the scarce cytoplasmic lipid bodies found in peritoneal mononuclear cells (**S3B Fig**) compared to elevated content within eosinophils (**Fig 4B**), and the lack of intracellular immunolabelling for newly synthesized LTC$_4$ (**Fig 4C**). Therefore, eosinophils seem to represent the cell source of PGD$_2$/DP2 receptor-driven cysLTs produced in the peritoneal compartment of *S. mansoni*-infected animals.

Moving from the peritoneal compartment to the granulomatous liver of *S. mansoni*-infected mice, CAY10471 produced an even more robust inhibition than those observed with HQL-79 treatment on the magnitude of eosinophilia within hepatic granulomas found 8 wpi (**Fig 5A**). More than simply ratifying HQL-79 data, and in clear contrast to data showing enhanced allergic pulmonary eosinophilia in DP2 deficient mice [63], CAY10471-driven findings upgraded DP2 to one of the key eosinophilotactic receptors in schistosomiasis, like CCR3 [51].

Besides inhibition of eosinophil accumulation, CAY10471 also promoted the reduction of total hepatic cysLTs (LTC$_4$/D$_4$/E$_4$) content within the *S. mansoni* infection-induced granuloma-enriched liver (**Fig 5A**). A positive correlation ($r^2$ = 0.7253; $p \leq 0.0018$) derived from these data indicates a product/producer relationship between cysLTs and granulomatous eosinophils under CAY10471 treatment (**Fig 5B**). Taking together with HQL-79 data (**Fig 3B**), we speculate that an *S. mansoni* infection-induced DP2 activation by PGD$_2$ stimulates *in situ* LTC$_4$ synthesis/release by eosinophils infiltrating hepatic granulomas, therefore bringing about the increased liver content of cysLTs.

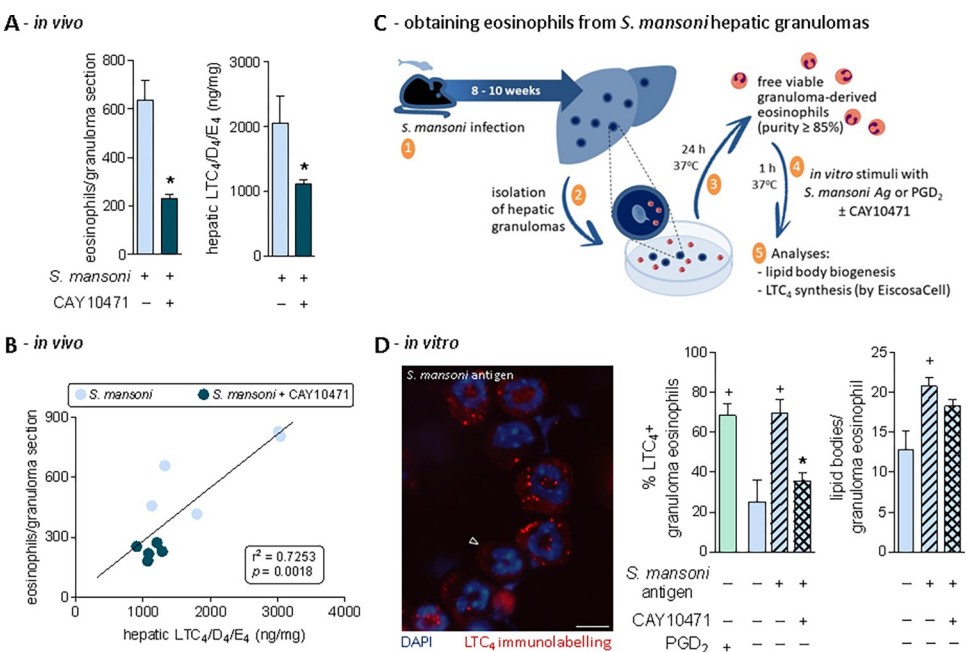

**Fig 5. DP2 receptor activation triggers *S. mansoni* granuloma eosinophil response: PGD$_2$-stimulated eosinophils as the cellular source of DP2-driven *de novo* synthesized LTC$_4$.** A shows enumeration of granuloma-infiltrating eosinophils as well as the release of cysLTs (LTC$_4$/D$_4$/E$_4$) detected by specific ELISA kits in liver homogenates. Values are expressed as the mean ± SEM from at least 5 animals *per* group (experiment was repeated at least once). $^+p < 0.05$ compared to non-infected control group. $^*p < 0.05$ compared to infected non-treated group. B displays linear regression curve illustrating the relationship between cysLTS levels and granulomatous eosinophilic reaction found in hepatic tissues. R-squared (r$^2$) and $p$ values for the linear regression are shown. Panel C schematizes protocol for the isolation of eosinophils from schistosomal hepatic granulomas (a cellular suspension displaying purity of about 85 to 90%). Image in D shows a representative image of intracellularly immunolabeled LTC$_4$ (in red; for EicosaCell preparation *vide* Methods) within granuloma-isolated eosinophils stimulated *in vitro* for 1 h with *S. mansoni* antigen (0.5 μg/mL). Cellular nuclei are labeled with DAPI (in blue; bar = 10 μm; arrowhead indicates an LTC$_4^-$ eosinophil). Graphs in D show the percentage of *in vitro* stimulated eosinophils (as indicated) exhibiting cytoplasmic immunostaining for LTC$_4$ (LTC$_4^+$ eosinophils) as well as the cytoplasmic numbers of eosinophil lipid body organelles (evaluated in osmium-stained cells). Values are expressed as the mean ± SEM from 5 preparations of granuloma-isolated eosinophils. $^+p < 0.05$ compared to non-stimulated eosinophils. $^*p < 0.05$ compared to *S. mansoni* antigen-stimulated eosinophils.

Despite the correlation between eosinophil infiltration and cysLT content in the liver, a demonstration that granulomatous eosinophils indeed generate LTC$_4$ was still to be determined. To investigate whether the eosinophils present in the fibrotic hepatic granulomas were LTC$_4$ synthesizing cells accountable for the increased cysLTs levels found in schistosomal livers, we isolated eosinophils from the *S. mansoni* hepatic granulomas, stimulated them *in vitro* with *S. mansoni* antigen or PGD$_2$ for 1 h (with or without CAY10471) and then analyzed lipid body biogenesis as well as intracellular LTC$_4$ synthesis (**Fig 5C**). In this *in vitro* experimental setting, direct stimulation with PGD$_2$ (25 nM) triggered rapid (1 h) intracellular LTC$_4$ synthesis within about 70% of granuloma-isolated eosinophils (**Fig 5D**). So, hepatic granulomatous eosinophils under PGD$_2$ stimulation are indeed capable of LTC$_4$ synthesis. Moreover, *in vitro* co-incubation of granuloma-isolated eosinophils with CAY10471 blocked *S. mansoni* antigen ability to induce LTC$_4$ synthesis (**Fig 5D**). So, at least *in vitro*, an autocrine/paracrine phenomenon takes place to regulate LTC$_4$ synthesis by *S. mansoni* granuloma-derived eosinophils. Concurring, rapid synthesis of PGD$_2$ (1 h) by granuloma-derived eosinophils is also triggered by *in vitro* stimulation with *S. mansoni* antigen (**S4 Fig**). It is noteworthy that all non-eosinophils cells present after processing of the granulomas were negative for PGD$_2$. Therefore, one

can speculate that, once eosinophils are within hepatic granulomatous environment 8 wpi, they may also assist stellate cells in synthesizing $PGD_2$.

For either human blood or in murine pleural fluid cells, eosinophil lipid bodies are the primary subcellular site of $PGD_2$-induced $LTC_4$ synthesis [11,12,64]. As shown in **Fig 5D**, immunolabelling of newly synthesized $LTC_4$ was in a punctate pattern, with cytoplasmic distribution apart from the perinuclear membrane and fully consistent in size and numbers with eosinophil lipid bodies. As confirmed here (**Fig 5D**, right graph), *in vitro* co-incubation with CAY10471 does not modify *S. mansoni* antigen-triggered lipid body biogenesis, a $PGD_2$-induced intracellular event downstream to DP1 receptor activation, rather than DP2 [15]. Of note, in response to $PGD_2$, eosinophils are the only characterized cells that synthesize $LTC_4$ within these cytoplasmic organelles [10].

## Activation of CysLT1 attenuates *S. mansoni* infection-driven granulomatous fibrosis

The two direct attempts to define eosinophil function in schistosomiasis employing mouse models of eosinophil lineage deficiency, including the studies by Swartz et al. (2006) and de Oliveira et al. (2022), did not uphold the original assumption of eosinophil effector cytotoxic function towards parasites. Instead, these pivotal studies have unveiled a more subtle immunomodulatory role for eosinophils in *S. mansoni*-evoked granulomatous pathogenesis [49,50]. Particularly regarding *S. mansoni* infection-triggered hepatic fibrosis within eosinophilic granulomas, de Oliveira and coauthors (2022) revealed that in the absence of eosinophils, reductions in TGF-β and IL-13 levels were accompanied by attenuation of hepatic fibrosis [49], thus indicating a pro-fibrogenic role for eosinophils in the disease. These findings are in clear contrast with the inverse relationship between *S. mansoni* infection-related eosinophilia and fibrogenesis observed here, whereby decreasing $PGD_2$-driven eosinophilia (**Figs 3** and **4**), HQL-79 and CAY-10471 did increase the fibrogenic cytokines as well as hepatic fibrosis (**Fig 2**). It seems that targeting specifically $PGD_2$-regulated eosinophil functions after the establishment of initial events of infection, rather than the entire eosinophil population from the beginning, appears to promote a distinct pattern of response in schistosomiasis.

We have identified at least one of these $PGD_2$/DP2-activated eosinophil effector functions that mediate the endogenous $PGD_2$/DP2-driven counter regulatory impact on fibrosis unveiled here: production of $LTC_4$ — a molecule with putative anti-fibrogenic effects on *S. mansoni* infection-induced hepatic granulomatous fibrosis [57]. To directly verify whether hepatic cysLTs down-modulate S. mansoni-induced granulomatous fibrosis, we employed the pharmacological blockade of cysLTs receptor CysLT1 by its selective antagonist, MK571; also delivered by osmotic pump system (flow rate of 2.6 µg/day; implanted at 3.5 wpi). As shown in **Fig 6**, even though the treatment with MK571 did not interfere with *S. mansoni* infection-induced $PGD_2$ production (**Fig 6A**) or eosinophil accumulation (**Fig 6B**) in hepatic granulomatous tissue 8 wpi, the antagonist did enhance the hepatic levels of pro-fibrogenic cytokines TGF-β and IL-13 (**Fig 6C**), without affecting the number of hepatic granulomas (**Fig 6D**). So, without interfering with the $LTC_4$ cell source or its molecular trigger, i.e. DP2-expressing eosinophils and $PGD_2$, the antagonism of CysLT1 receptors reproduced the HQL-79 or CAY10471 pro-fibrogenic effect. This final finding indicates that activation of CysLT1 by eosinophil-derived $LTC_4$ and its active metabolites within the schistosomal granuloma represents a mechanism by which endogenous $PGD_2$/DP2 axes restrains the buildup of fibrosis through hepatic parenchyma of infected animals.

Therefore, as schematized in **Fig 7**, our data uncover an endogenous $PGD_2$-driven counter regulatory pathway, which intrinsically limits the development of schistosomal hepatic fibrosis.

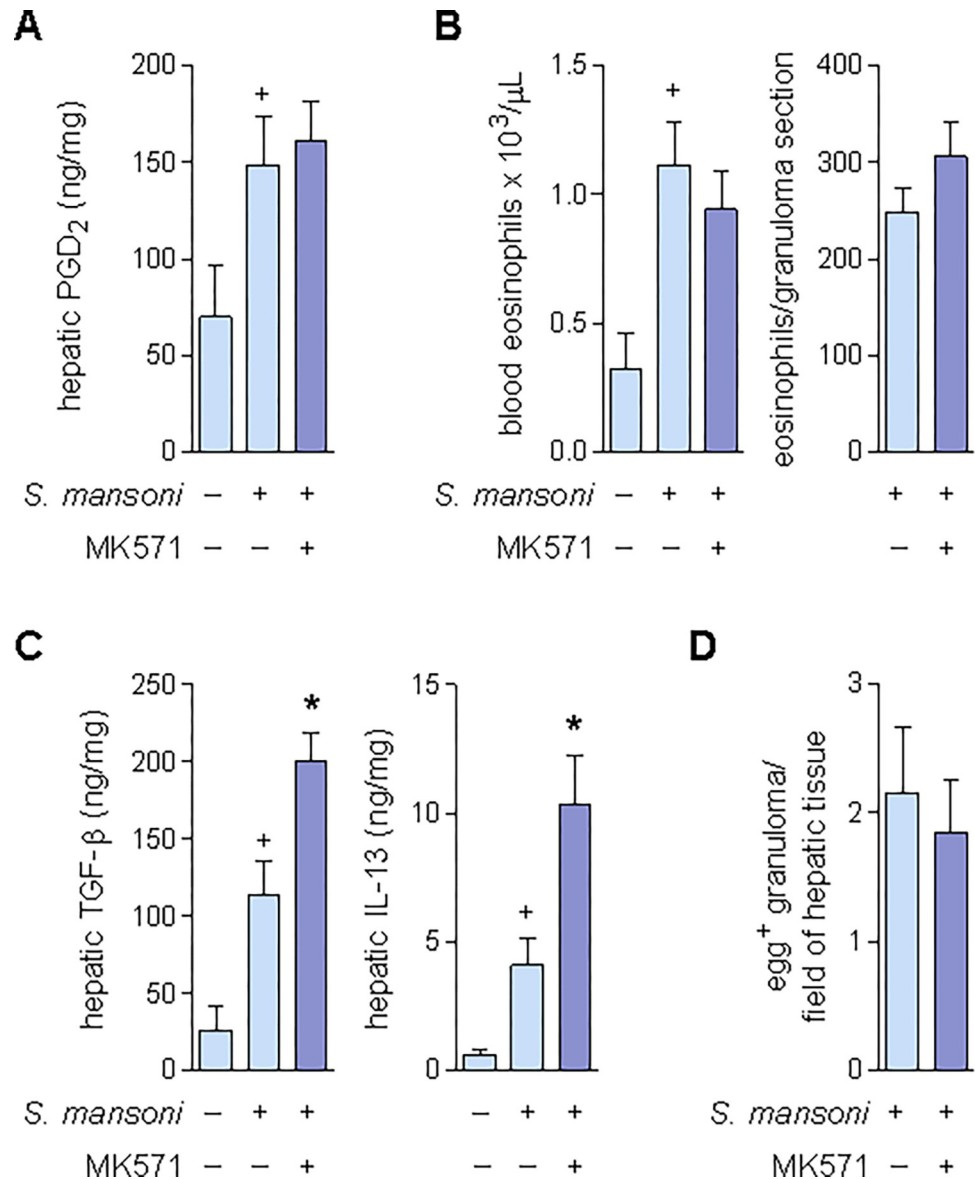

**Fig 6. MK571 potentiates fibrogenic reaction within *S. mansoni* infection-driven hepatic granuloma, without interfering with PGD$_2$ synthesis or the eosinophilic reaction.** Continuous treatment (delivered by implanted osmotic pumps) with MK571 (2.6 μg/day) was initiated at 3.5 wpi with *S. mansoni*. Blood samples and livers were collected 8 wpi. **A** shows PGD$_2$ levels in liver homogenates detected by specific EIA kits. **B** shows eosinophil numbers found in either peripheral blood or hepatic granulomas. **C** shows levels of TGF-β and IL-13 detected by specific ELISA kits in liver homogenates. **D** shows total numbers of egg-encasing hepatic granulomas. Values are expressed as the mean ± SEM from at least 7 animals *per* group. $^+p < 0.05$ compared to non-infected control group. $^*p < 0.05$ compared to infected non-treated group.

Stepwise, by the onset of hepatic granulomatous reaction formed around *S. mansoni* eggs, hPGDS-expressing structural stellate cells of schistosomal granulomas synthesize PGD$_2$. Via activation of DP2 receptors, stellate cells-derived PGD$_2$ both chemoattracts and activates eosinophils to synthesize/release LTC$_4$. Of note, by 8 wpi, PGD$_2$ synthesis may also be performed by newly arrived eosinophils, which autocrinally stimulate LTC$_4$ release. Acting on CysLT1 receptors-expressing cells, intragranulomatous LTC$_4$/D$_4$/E$_4$ decrease the fibrogenic

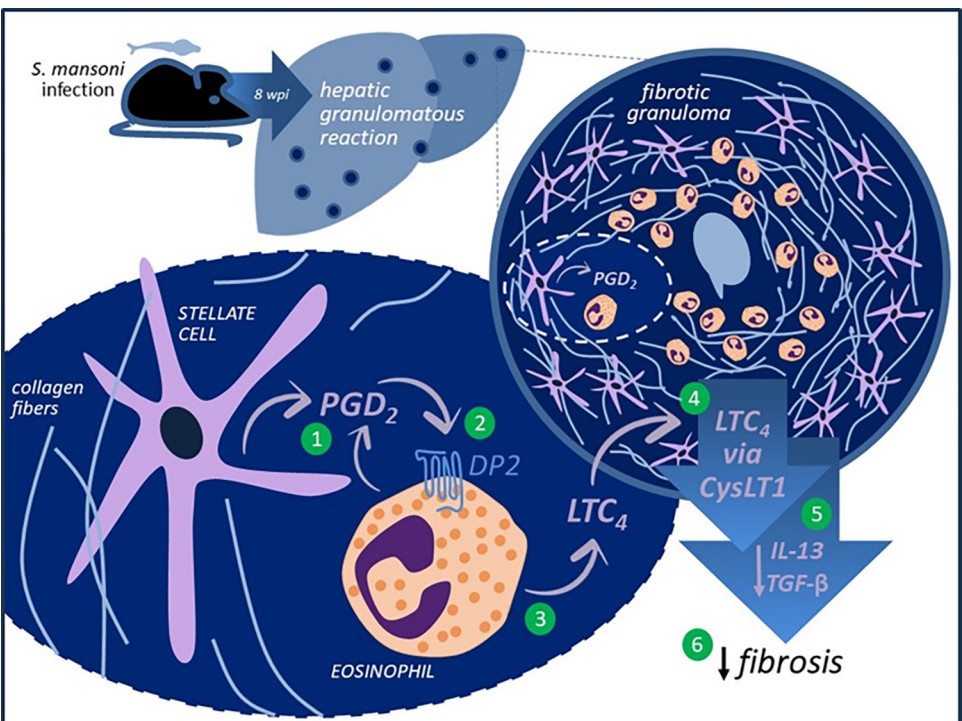

**Fig 7. Endogenous PGD$_2$/DP2-driven counter regulatory impact on the development of schistosomal hepatic fibrosis.** The induction of liver granulomatous reaction in an experimental mouse model of *S. mansoni* infection, is characterized by an initial synthesis of PGD$_2$ by hPGDS-expressing hepatic stellate cells (after later establishment of granulomatous eosinophilia, eosinophils contribute to PGD$_2$ production) (1). Activation of DP2 receptors in granulomatous eosinophils (2) triggers the induction of LTC$_4$ synthesis (3). Released LTC$_4$ and/or its metabolites activate CysLT1 receptors-expressing cells (4), promoting the reduction of fibrogenic cytokines TGF-β and IL-13 (5) that culminates with counter regulation of the development of hepatic granulomatous fibrosis around the *S. mansoni* egg (6).

cytokines TGF-β and IL-13, attenuating the development of hepatic granulomatous fibrosis around the *S. mansoni* egg.

## Conclusions

The detrimental effect of both HQL-79 and CAY10471 on *S. mansoni* infection-driven hepatic granulomatous inflammation uncovered here has disclosed an unanticipated endogenous protective mechanism of the hepatic tissue — the *in vivo* anti-fibrogenic activity of hPGDS-derived PGD$_2$ via DP2 activation. In addition, our pharmacological approaches have also unveiled a potential novel PGD$_2$-driven regulatory mechanism: DP2 receptors-elicited hepatic granulomatous eosinophils as a major cellular source of anti-fibrogenic LTC$_4$ in *S. mansoni* infection. Noteworthy, since LTC$_4$ is known to function also as an intracrine signal capable of triggering IL-4 secretion from eosinophils [65,66], cysLTs may emerge as immunoregulators of the IL-4-mediated type 2 immune response of schistosomiasis.

A main driver for the current study was the *in vitro* data on PGD$_2$ role in hepatic tissue-related fibrosis suggesting that PGD$_2$ displays deleterious functions in the context of *S. mansoni* infection [8,21]; and more importantly, the resulting recommendations for PGD$_2$-targeting therapeutic maneuvers in schistosomiasis [35–37]. Because presently there are other clinically approved uses for DP2 receptor antagonism in type 2 immune response-biased pathologies [i.e., ramatroban (Baynas) in allergic rhinitis], our current study compels that drug

repurposing should not arise directly from the *in vitro* studies, and most importantly should be avoided in this case. Concurring, the mechanism involved in DP2 receptor anti-fibrogenic function has recently emerged as a far more complex activity. For instance, it has been shown that the DP2 molecule appears to traffic back and anchor to endoplasmic reticulum membranes where, in a fashion non-related to PGD$_2$, DP2 evokes degradation of collagen mRNA and decreases intracellular collagen biosynthesis [67]. Also, data derived from genetically deficient animals should be carefully analyzed before being incorporated into treatment strategies as they may not reflect the proper window of opportunity for the infected patient at risk of developing complications due to hepatic fibrosis. Indeed, based on our current *in vivo* findings, promoting pharmacological DP2 selective agonism (not antagonism) may be the right track to achieve anti-fibrogenic effects.

## Materials and methods

### Ethics statement

Male C57BL/6 mice weighing 20 to 25 g were obtained from either CCS/UFRJ or FIOCRUZ breeding centers, raised, and maintained under the same housing conditions. Animals were housed in a temperature-controlled room, under a 12:12 h light cycle with access to filtered water and chow *ad libitum*. All animal care and experimental protocols were conducted following the guidelines of the Brazilian Council for Care and Use of Experimentation Animals (CONCEA) in accordance with Brazilian Federal Law number 11.794/2008, which regulates the scientific use of animals (license numbers CEUA115/14 and CEUA139/21 by the Committee for Ethics in Animal Experimentation of Federal University of Rio de Janeiro).

### *S. mansoni* infection and treatments

Mice were infected by active percutaneous penetration of *S. mansoni* infective stage (BH strain; Institute Oswaldo Cruz, FIOCRUZ, RJ), by exposition of washed mouse paws to 60 alive cercariae for 40 min. Uninfected age-matched mice were used as controls. At 3.5 weeks post infection (wpi), osmotic pumps (Alzet pump; flow rate of 0.11 µL/hour — as stated by the manufacturer) — containing 100 µL of either HQL-79 (1 mg/mL), CAY10471 (670 µg/mL), or MK571 (1 mg/mL) solutions producing a cumulative dose of 2.6, 1.7, or 2.6 µg/day, respectively — were implanted subcutaneously in infected and non-infected mice. All three drugs were diluted with 0.1% DMSO in sterile saline. Osmotic pumps containing vehicle solution implanted subcutaneously did not cause changes in mice survival rate, hepatic dysfunction, or inflammatory alterations within implantation sites, peripheral blood or peritoneal cavities of either non-infected or infected mice (S1 Table). All animals were euthanized under anesthesia after 3, 6, or 8 wpi and during this time, they were maintained under the same housing conditions described before.

### Analysis of parasitological parameters

To ascertain the establishment of the infection and characterize oviposition onset, the presence of *S. mansoni* eggs in feces and hepatic tissue were determined. *S. mansoni* eggs were identified by morphology (containing sharp lateral spines) and enumerated by copro-parasitological thick-smear Kato-Katz method (Helm-Test Biomanguinhos; FIOCRUZ), while the presence of *S. mansoni* eggs found trapped in hepatic granulomas was analyzed in liver histological preparations (as described below).

## Histopathological analysis

Liver samples were washed in saline, fixed in 10% buffered formalin, dehydrated in alcohol, and embedded in paraffin. Sections (5 μm of thickness) were stained with (i) hematoxylin-eosin for enumeration and area analysis of granulomas with entrapped egg; (ii) modified PicroSirius technique for collagen fibers deposition detection; or (iii) Sirius Red for granuloma-infiltrating eosinophils quantification. Acquisition of digital photographs and quantitative analysis of hepatic tissue sections were carried out using a slider scanner (Pannoramic Midi 3DHistec or Motic Easy Scan) and a computer-assisted image analyzer (Slide Viewer 2.4 or Qu Path 0.4.3). All evaluations were performed by two different blinded observers.

Under light microscopy, the areas of hepatic granulomas (20 granulomas *per* animal) were determined in digital images (acquired at 20x magnification) of histological sections, containing central eggs, randomly chosen, using the measurement tool of the image analyzer. Egg-encasing granulomas counting was carried out at a low power (20x) magnification under light microscopy. The mean number of egg-encasing granulomas *per* field was calculated for each infected mouse. By confocal microscopy, total fluorescent intensity within digital images of individual granuloma areas (10 granulomas *per* animal) was quantified using Image J software. All eosinophils identified in individual granuloma areas were counted in digital images acquired under 80x magnification light microscopy. Ten random granulomas *per* animal were analyzed and results were expressed as eosinophils/granuloma.

## Peripheral blood, peritoneal, and bone marrow eosinophilia

Blood eosinophilia was analyzed by light microscopy of blood smears stained with Panoptic kit. To evaluate peritoneal and bone marrow eosinophilia, cells from peritoneal cavities and bone marrows of removed femurs were harvested with RPMI 1640 medium (Sigma) and cytospun towards slides. Total leukocyte enumeration was performed in Neubauer chambers and differential eosinophil count in Panoptic kit-stained slides under light microscopy.

## Eosinophils isolation from *S. mansoni*-driven hepatic granuloma

Mice livers recovered from 8 to 10 wpi with *S. mansoni* were homogenized in RPMI 1640 medium. Intact hepatic granulomas were allowed to sediment, washed 3 times with RPMI, and then incubated overnight (37˚C; 5% $CO_2$). In culture bottles, granuloma-released eosinophils are non-adherent cells in suspension, while other cells are found still within granulomas or attached to the plastic. Recovered cells were analyzed under light microscopy after Panoptic kit staining to determine the percentage of eosinophils in suspension. The eosinophil fraction used in *in vitro* experiments was composed by 80 to 90% purified eosinophils.

Isolated eosinophils ($3 \times 10^6$ cells/mL) were co-incubated or not with CAY10471 (200 nM; Cayman Chemicals) and then stimulated for 1 h (37˚C; 5% $CO_2$) with *S. mansoni* antigen (0.5 μg/mL; Cusabio) or $PGD_2$ (25 nM; Cayman Chemicals). After incubation, eosinophil samples were promptly fixed with PFO for osmium staining or placed in EDAC solution for EicosaCell assay (see below).

## Evaluation of hepatic collagen synthesis

As an indirect quantitative assay determining amounts of collagen molecules, hepatic fibrosis was also evaluated by measuring hydroxyproline levels in liver homogenates. Hydroxyproline content was determined in dehydrated/hydrolyzed liver fragments by a colorimetric method in chloramine-T buffer (Sigma, USA) with Ehrlich's reagent (Sigma)/perchloric acid (Merck)

and absorbance was measured at 557 nm. Results were expressed as ng of hydroxyproline *per* mg of hepatic tissue.

## Analysis of lipid mediators production

Amounts of eicosanoids $PGD_2$, $PGE_2$, and cysteinyl leukotrienes ($LTC_4/D_4/E_4$) found in peritoneal fluid or homogenized liver fragments were measured by specific commercial EIA kits, according to the manufacturer's instructions (Cayman Chemicals).

For cysLTs production, intracellular $LTC_4$ synthesis was alternatively analyzed by the EicosaCell methodology [68] — an immuno-assay that immobilizes the newly formed eicosanoid within active synthesizing cells. Briefly, peritoneal cells or *in vitro*-stimulated granuloma-derived eosinophils (cell suspensions in RPMI) were immediately mixed with an equal volume of a 0.2% solution of 1-ethyl-3-(3-dimethylamino-propyl) carbodiimide (EDAC; in PBS), used to crosslink eicosanoid carboxyl groups to amines in proteins. After a 10 min incubation at room temperature with EDAC, eosinophils were washed with PBS, cytospun onto glass slides, fixed with PFO (2%), incubated with PBS with 1% BSA for 15 min, and then incubated with a rabbit anti-$LTC_4$ antibody (Cayman Chemicals) overnight. The cells were washed with PBS containing 1% BSA (3 times 10 min) and then incubated with Alexa594 donkey anti-rabbit secondary IgG (Jackson) for 1 h. Nuclear visualization by DAPI staining was employed to distinguish polymorphonuclear eosinophils from mononuclear cells. EicosaCell images were obtained using an Olympus BX51 fluorescence microscope, equipped with a 100X objective in conjunction with LAS-AF 2.2.0 Software.

## Cytokines measurements

Amounts of TGF-β, IL-5, and IL-13 in liver fragments or in serum samples were measured by commercial ELISA kits, according to the manufacturer's instructions (R&D Systems and/or Peprotech).

## Lipid body staining and enumeration

For lipid body counting, cells in cytospin slides were fixed in 3.7% paraformaldehyde and stained with 1.5% OsO4 in 0.1 M cacodylate buffer, as previously described [64]. By bright field microscopy, fifty consecutively eosinophils/slide were evaluated in a blinded fashion by more than one observer.

## Statistical analysis

Results are expressed as means ± SEM (standard error of the mean) and were analyzed by one-way ANOVA, followed by Student-Newman-Keuls test or by Student's *t* test, using Prism software (GraphPad Software, Inc., San Diego, CA, USA). Differences were considered significant when $p < 0.05$. Two independent experiments were performed for each $PGD_2$-targeting treatment studied: HQL-79 and CAY10471.

## Supporting information

**S1 Fig. Inhibition of $PGD_2$ synthesis by HQL-79 treatment did not affect *S. mansoni* infection-induced immune polarization to a type 2-biased response in mice. A** shows serum levels of IL-5 and IL-13. **B** shows eosinophil numbers found at bone marrow. Values are expressed as the mean ± SEM from at least 5 animals *per* group (experiment was repeated at least once). [+]$p < 0.05$ compared to non-infected control group.
(TIF)

**S2 Fig. PGD$_2$ synthesis within liver of *S. mansoni* infected mice precedes installation of hepatic granulomatous eosinophilia: hepatic stellate cells as PGD$_2$ cellular source.** Peritoneal lavages and livers were collected 6 wpi with *S. mansoni* cercariae. **A** shows PGD$_2$ amounts detected by specific EIA kit in cell-free peritoneal fluids and in liver homogenates. In **B**, total numbers of egg-encasing hepatic granulomas are shown. **C** shows eosinophil numbers found in either peritoneal compartment, hepatic granuloma, peripheral blood, or bone marrow space. **D** shows peritoneal and liver LTC$_4$ levels. Values are expressed as the mean ± SEM from at least 5 animals *per* group. $^+p < 0.05$ compared to non-infected control group. In **E**, hepatic granuloma-derived stellate cells were isolated 6 wpi from *S. mansoni*-infected mice and then stimulated *in vitro* for 2 h with total lipids extracted from isolated parasites; PGD$_2$ synthesis was quantified in cell-free supernatants. Values are expressed as the mean ± SEM from 2 preparations of hepatic granuloma-derived stellate cells.
(TIF)

**S3 Fig. Antagonism of DP2 receptor by CAY10471 treatment did not affect *S. mansoni* infection-elicited population of peritoneal mononuclear cells. A** shows numbers of mononuclear cells found at peritoneal space as well as representative images of peritoneal cells 8 wpi of *S. mansoni*-infected mice (top image) and in mice treated with CAY10471 and *S. mansoni* infected (bottom image). **B** shows numbers of cytoplasmic lipid body organelles found in peritoneal mononuclear cells. Values are expressed as the mean ± SEM from at least 5 animals *per* group (experiment was repeated at least once). $^+p < 0.05$ compared to non-infected control group.
(TIF)

**S4 Fig. *In vitro* stimulated hepatic granuloma-isolated eosinophils synthesize PGD$_2$.** Eosinophils were isolated from 8 wpi schistosomal hepatic granulomas (purity about 90%) and stimulated *in vitro* for 1 h with *S. mansoni* antigen (0.5 μg/mL). Percentage of *in vitro* stimulated eosinophils (as indicated) exhibiting cytoplasmic immunostaining for LTC$_4$ (LTC$_4$$^+$ eosinophils) under fluorescence microscopy in Eicosacell preparations are shown. Values are expressed as the mean ± SEM from 3 preparations of granuloma-isolated eosinophils. $^+p < 0.05$ compared to non-stimulated eosinophils.
(TIF)

**S1 Table. Lack of impact of 0.1% DMSO solution (the vehicle solution employed for HQL-79, CAY10471 or MK571 treatments delivered by osmotic pumps) on non-infected or S. mansoni-infected animals.** S. mansoni infection in mice was achieved by active percutaneous penetration by 60 cercariae. Osmotic pumps containing a 0.1% DMSO solution were subcutaneously implanted (3.5 wpi) in both non-infected and S. mansoni-infected animals (columns labelled "DMSO"). Animals that were not implanted with subcutaneous pumps were used as controls (columns labelled "-"). All animals were alive at 8 wpi, when they were euthanized. Livers of all non-infected animals were normal, showing no macroscopic or histopathological alterations. Individual values and mean ± SEM from 2 or 3 animals per group are displayed.
(DOCX)

**S2 Table. Raw data for the graphs.** The data for each group presented in the manuscript is listed in individual tables for each graph indicated by Figure number and letter. Mean ± SEM for each group is provided in the bottom line.
(DOCX)

## Author Contributions

**Conceptualization:** Giovanna N. Pezzella-Ferreira, Camila R. R. Pão, Isaac Bellas, Tatiana Luna-Gomes, Valdirene S. Muniz, Natalia R. T. Amorim, Claudio Canetti, Patricia T. Bozza, Bruno L. Diaz, Christianne Bandeira-Melo.

**Data curation:** Bruno L. Diaz, Christianne Bandeira-Melo.

**Formal analysis:** Giovanna N. Pezzella-Ferreira, Camila R. R. Pão, Isaac Bellas, Tatiana Luna-Gomes, Valdirene S. Muniz, Ligia A. Paiva, Natalia R. T. Amorim, Claudio Canetti, Patricia T. Bozza, Bruno L. Diaz, Christianne Bandeira-Melo.

**Funding acquisition:** Christianne Bandeira-Melo.

**Investigation:** Giovanna N. Pezzella-Ferreira, Camila R. R. Pão, Isaac Bellas, Tatiana Luna-Gomes, Valdirene S. Muniz, Ligia A. Paiva, Natalia R. T. Amorim, Claudio Canetti, Patricia T. Bozza, Bruno L. Diaz, Christianne Bandeira-Melo.

**Methodology:** Giovanna N. Pezzella-Ferreira, Camila R. R. Pão, Isaac Bellas, Tatiana Luna-Gomes, Valdirene S. Muniz, Ligia A. Paiva, Natalia R. T. Amorim, Claudio Canetti, Patricia T. Bozza, Bruno L. Diaz, Christianne Bandeira-Melo.

**Project administration:** Bruno L. Diaz, Christianne Bandeira-Melo.

**Resources:** Christianne Bandeira-Melo.

**Supervision:** Bruno L. Diaz, Christianne Bandeira-Melo.

**Writing – original draft:** Bruno L. Diaz, Christianne Bandeira-Melo.

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
