## [Decision Letter · Decision Letter 0]

4 Jan 2024

Dear Dr. Diaz,

Thank you very much for submitting your manuscript "Host PGD2 acting on DP2 receptor attenuates Schistosoma mansoni infection-driven hepatic granulomatous fibrosis" for consideration at PLOS Pathogens. As with all papers reviewed by the journal, your manuscript was reviewed by members of the editorial board and by several independent reviewers. In light of the reviews (below this email), we would like to invite the resubmission of a significantly-revised version that takes into account the reviewers' comments, including performing the additional experiments to target eosinophils and PGD2/DP2.

We cannot make any decision about publication until we have seen the revised manuscript and your response to the reviewers' comments. Your revised manuscript is also likely to be sent to reviewers for further evaluation.

Sincerely,

Meera Goh Nair

Academic Editor

PLOS Pathogens

Dominique Soldati-Favre

Section Editor

PLOS Pathogens

Kasturi Haldar

Editor-in-Chief

PLOS Pathogens

orcid.org/0000-0001-5065-158X

Michael Malim

Editor-in-Chief

PLOS Pathogens

orcid.org/0000-0002-7699-2064

Please address all reviewers' comments including performing the additional experiments to target eosinophils and PGD2/DP2

Reviewer's Responses to Questions

**Part I - Summary**

Reviewer #1: The authors of this study demonstrated that the activation of PGD2 and its receptor may play a protective role in liver fibrosis following S. mansoni infection. Key findings from this research reveal that inhibiting PGD2 synthesis using HQL-79 and blocking the DP2 receptor with an antagonist resulted in elevated levels of TGFb and IL-13 in the liver, which are believed to promote fibrogenesis. An intriguing discovery in this study is that treatment with CAY10471, a known DP2 receptor blocker, led to a decrease in the number of eosinophils in the liver sections of S. mansoni-infected mice. Additionally, CAY10471 treatment also promoted a fibrotic reaction within the hepatic tissue of infected mice. The authors further established a link between LTC4 production and liver fibrosis, identifying eosinophils as the source of LTC4. Both in vitro and in vivo evidence demonstrated reduced levels of LTC4 when mice or eosinophils were treated with CAY10471. The conclusion that PGD2 activates DP2 receptors, stimulating cysLTs production by eosinophils, and subsequently inhibiting liver fibrosis is both interesting and significant. The paper is well-written and deserves publication.

Reviewer #2: (No Response)

Reviewer #3: The liver is the main cause of health problems associated with schistosomiasis. A potential role for PGD2 in promoting liver fibrosis has been demonstrated in vitro. Thus, it was suggested that PGD2 inhibition would benefit people with schistosomiasis. However, there is no direct evidence in a schistosomiasis mouse model.

The Authors describe the effect of inhibiting the production or action of PGD2 in a mouse model of schistosomiasis. They also investigated how this PGD2 effect is linked to eosinophils and to LTC4. Surprisingly the data indicate that PGD2 protects the liver from fibrosis rather than causing it. Thus, its inhibition might aggravate liver failure and not benefit patients with schistosomiasis.

The paper's background is well reasoned and brings the rationale to assess the role of PGD2 in liver fibrosis in Schistosoma mansoni infected mice. The subject is important. The research is exacting, well-executed and well-documented.

However, the manuscript is too complex. I would have suggested that they go more in depth on one of the various directions: which cells besides eosinophils are responsible for the PGD2 pro-fibrotic or anti-fibrotic effects, by which mechanisms in vitro and in vivo is this carried out, etc. For example, the Authors mention the important role of mast cells as main cells producing PGD2 and being present in the liver granulomas but fail to study this.

**Part II – Major Issues: Key Experiments Required for Acceptance**

Reviewer #1: However, there are a few weak points that need to be addressed:

1. It is important to discuss the potential differences in the consequences of DP2 deficiency/blockade on eosinophils in various models. For example, in an allergic inflammatory model of asthma, DP2/CRTH2 knockout mice displayed enhanced eosinophil recruitment into the lung compared to wild-type littermate mice (The Journal of Immunology, 175(4), 2056-2060). Similarly, in cases of renal fibrosis and lung fibrosis, DP2/CRTH2 has been shown to have profibrotic roles (American Journal of Respiratory Cell and Molecular Biology, 67(2), 201-214).

2. It would be beneficial to include in vivo models with DP2/CRTH2 conditional deletion in eosinophils to further elucidate how PGD2/DP2 contributes to liver pathology during S. mansoni infection.

3. If establishing an eosinophil-specific knockout of DP2 is not feasible, the authors should consider supplementing their data to further support the contribution of eosinophils in alleviating liver fibrosis caused by S. mansoni infection, specifically through DP2 and LTC4 production. One possible approach could be to investigate the effects of depleting eosinophils. By depleting eosinophils in an animal model, the authors could observe the impact on liver fibrosis progression and evaluate whether the absence of eosinophils alters the protective effects mediated by the PGD2-DP2-LTC4. This additional data would strengthen the understanding of the role of eosinophils in the context of liver fibrosis and help solidify the conclusions drawn in the study.

Reviewer #2: (No Response)

Reviewer #3: To strengthen the paper, additional experiments could be conducted on:

-- Eosinophil deficient mice in which PGD2 activity is inhibited

-- PGD2 knock-out mice

-- In mice in which pharmacological blockage of LOX5 is achieved

**Part III – Minor Issues: Editorial and Data Presentation Modifications**

Reviewer #1: (No Response)

Reviewer #2: (No Response)

Reviewer #3: The manuscript is too heavy and complex. It is not fluid reading. The itemization -- 1,2,3,4 -- is too much.

It should be re-written and shortened to make it easier to read and understand.

PLOS authors have the option to publish the peer review history of their article (what does this mean?). If published, this will include your full peer review and any attached files.

Reviewer #1: No

Reviewer #2: No

Reviewer #3: No
---

## [Decision Letter · Decision Letter 1]

19 Jul 2024

Dear Dr. Diaz,

Thank you very much for submitting your manuscript "Endogenous PGD2 acting on DP2 receptor counter regulates Schistosoma mansoni infection-driven hepatic granulomatous fibrosis" for consideration at PLOS Pathogens. As with all papers reviewed by the journal, your manuscript was reviewed by members of the editorial board and by several independent reviewers. The reviewers appreciated the attention to an important topic. Based on the reviews, we are likely to accept this manuscript for publication, providing that you modify the manuscript according to the review recommendations.

There are two outstanding issues raised by reviewer # 4 that shown require your attention  and a mini-revision  prior to acceptance.

Sincerely,

Dominique Soldati-Favre

Section Editor

PLOS Pathogens

Dominique Soldati-Favre

Section Editor

PLOS Pathogens

Michael Malim

Editor-in-Chief

PLOS Pathogens

orcid.org/0000-0002-7699-2064

Reviewer Comments (if any, and for reference):

Reviewer's Responses to Questions

**Part I - Summary**

Reviewer #1: The data from additional experiments in the revised manuscript addressed my concerns despite the fact that the eosinophil specific genetic mouse models are not available. The descriptions to put less emphasis on eosinophil specific roles as shown in line 107 to 113, Page 71 were appropriate.

Reviewer #4: The authors demonstrated that PGD2 by activating DP2 receptors stimulates cysLTs production by eosinophils, while endogenously down-regulates the hepatic fibrogenic process of S. mansoni granulomatous reaction – an in vivo protective function which demands caution in the future therapeutic attempts in targeting PGD 2/DP2 in schistosomiasis. The authors inhibited the production or action of PGD2, and antagonized CysLT1 in a mouse model of schistosomiasis. Furthermore, in vitro experiments provided further evidence for the presented conclusions.

The study is well conducted and the data presented represent a valuable contribution to research into schistosomiasis.

**Part II – Major Issues: Key Experiments Required for Acceptance**

Reviewer #1: none

Reviewer #4: Figures 2 A and B: Hepatic Hydroxyproline levels should be checked carefully. There seems to be an error in the presented data. 100-180 µg/g hydroxyproline can be found in the liver of healthy C57BL/6 mice while 300-400µg hydroxyproline per g liver tissue was detected in mice infected with S. mansoni for 9 weeks (PMID: 37990129).

**Part III – Minor Issues: Editorial and Data Presentation Modifications**

Reviewer #1: (No Response)

Reviewer #4: 1. Complicated wording makes understanding difficult:

e.g. Lines 298-302: First, distinct from the eosinophilic reaction impairment observed in DP1

receptor-deficient model [19] , HQL-79-unveiled PGD2 role in S. mansoni infection

driven eosinophilia does not correspond to an inevitable consequence of a large

inhibition of S. mansoni parasitism or overall lack of subsequent type 2 immune

response.

2. Perhaps separating the results and discussion into two separate sections would make the manuscript easier to understand.

PLOS authors have the option to publish the peer review history of their article (what does this mean?). If published, this will include your full peer review and any attached files.

Reviewer #1: No

Reviewer #4: No

Figure Files:

Data Requirements:

Reproducibility:

References:

---

## [Editor Report · Decision Letter 2]

7 Aug 2024

Dear Dr. Diaz,

We are pleased to inform you that your manuscript 'Endogenous PGD2 acting on DP2 receptor counter regulates Schistosoma mansoni infection-driven hepatic granulomatous fibrosis' has been provisionally accepted for publication in PLOS Pathogens.

Best regards,

Dominique Soldati-Favre

Section Editor

PLOS Pathogens

Dominique Soldati-Favre

Section Editor

PLOS Pathogens

Michael Malim

Editor-in-Chief

PLOS Pathogens

orcid.org/0000-0002-7699-2064
---

## [Editor Report · Acceptance letter]

15 Aug 2024

Dear Dr. Diaz,

We are delighted to inform you that your manuscript, "Endogenous PGD2 acting on DP2 receptor counter regulates Schistosoma mansoni infection-driven hepatic granulomatous fibrosis," has been formally accepted for publication in PLOS Pathogens.

Best regards,

Michael Malim

Editor-in-Chief

PLOS Pathogens

orcid.org/0000-0002-7699-2064